# Recent Progress on Green New Phase Extraction and Preparation of Polyphenols in Edible Oil

**DOI:** 10.3390/molecules28248150

**Published:** 2023-12-18

**Authors:** Feng Liang, Xue Li, Yu Zhang, Yi Wu, Kaiwen Bai, Romero Agusti, Ali Soleimani, Wei Wang, Shumin Yi

**Affiliations:** 1College of Biological and Chemical Engineering, Zhejiang University of Science and Technology, Hangzhou 310023, China; liangfeng1996@126.com (F.L.); mbb_wuyi@163.com (Y.W.); kaiwen_bai@163.com (K.B.); 2Institute of Agro-Product Safety and Nutrition, Zhejiang Academy of Agricultural Sciences, Hangzhou 310021, China; hei.semeng@163.com (X.L.); zhyu7711@gmail.com (Y.Z.); 3Institute of Agriculture and Food Research and Technology, Reus, El Morell Road, 43120 Constantí, Spain; agusti.romero@irta.cat; 4Department of Horticulture, Faculty of Agriculture, University of Zanjan, Zanjan 45371-38791, Iran; asoleimani@znu.ac.ir; 5School of Food Science and Engineering, Bohai University, Jinzhou 121013, China

**Keywords:** edible oil, polyphenols, green extraction, green solvent

## Abstract

With the proposal of replacing toxic solvents with non-toxic solvents in the concept of green chemistry, the development and utilization of new green extraction techniques have become a research hotspot. Phenolic compounds in edible oils have good antioxidant activity, but due to their low content and complex matrix, it is difficult to achieve a high extraction rate in a green and efficient way. This paper reviews the current research status of novel extraction materials in solid-phase extraction, including carbon nanotubes, graphene and metal–organic frameworks, as well as the application of green chemical materials in liquid-phase extraction, including deep eutectic solvents, ionic liquids, supercritical fluids and supramolecular solvents. The aim is to provide a more specific reference for realizing the green and efficient extraction of polyphenolic compounds from edible oils, as well as another possibility for the future research trend of green extraction technology.

## 1. Introduction

As the standard of living improves day by day, people pay more and more attention to the nutritional matching of daily meals, and fats and oils can provide a large amount of high-quality proteins and are rich in nutrients and are easy to be absorbed, so they naturally become one of the indispensable nutrients in the population’s diet. With the increase in the types, production and consumption of edible oils, the selection of high-quality edible oils with high nutritional value has received more and more attention [1].

Edible oils contain some micronutrients also known as lipid concomitants, such as phenolic compounds, phytosterols, squalene and fat-soluble vitamins, etc., which are very low in content but play a vital role in human health. Among them, polyphenolic compounds are a class of plant secondary metabolites; different types of edible oils contain different types and contents of polyphenolic compounds [2], for example, avocado phenol in olive oil, and peach kernel oil triphenylene is a common polyphenolic compounds. These dispersed polyphenols can be found to be made up of multiple benzene ring structures and multiple hydroxyl groups, which include phenolic acids, tannins, isoflavones and stilbenes, etc. [3,4]. And polyphenolic compounds have a variety of pharmacological applications, including antibacterial and anti-inflammatory [5], antioxidant, diabetes prevention, cardioprotective and tumor-growth-inhibiting properties [6]. Consumption of olive oil rich in phenolic compounds in the diet of Mediterranean populations has been reported to have cardioprotective effects, while showing favorable effects on the regulation of obesity [7].

Phenolic compounds are important components that occupy a major role in most functional foods and nutraceuticals. However, there are still many problems to overcome in the extraction and detection of phenolic compounds in edible oils [8]. First, the problem of extraction rate. Due to the low content of polyphenolic compounds in edible oils and the complexity of their matrix, many extraction methods are unable to extract them at the maximum yield level. Secondly, environmental issues. Conventional extraction techniques require the use of large amounts of organic solvents in the extraction of polyphenolic compounds and have the disadvantages of long extraction time, low selectivity and low extraction efficiency. Therefore, there is a need to find sample pretreatment techniques and extraction methods that can be more green and efficient, which need to ensure that the extraction method is easy to operate, shorten the extraction time, and reduce the use of organic solvents, etc. [9]. Green extraction technology is an environmentally friendly extraction technology developed with green chemistry as the core. Its sample pretreatment technology is to replace toxic solvents with non-toxic solvents, through a variety of auxiliary means, such as ultrasound-assisted technology, ultrahigh-pressure-assisted technology and microwave-assisted technology, etc. so that the extraction technology can overcome the limitations of the traditional method of sample preparation, thereby improving extraction efficiency and selectivity [10].

Based on the concept of green chemistry, this paper discusses in detail the green extraction technology of polyphenolic compounds in edible oils and related novel green chemistry materials. The contents include the application of carbon nanotubes, graphene and metal–organic frameworks involved in novel extraction materials in solid-phase extraction, and the application of deep eutectic solvents, ionic liquids, supercritical fluids and supramolecular solvents in liquid-phase extraction. Emphasis is placed on the definition and classification of novel extraction materials and green reagents, and their application in natural product extraction.

## 2. Research Progress on Green Extraction Techniques

In recent years, with the booming development of green chemistry, more and more types of green solvents in experimental application, and the development of each new green solvent, must be based on its own characteristics, through the selection of resources from the use of alternative organic solvents, the reduction of energy consumption, and the reduction of the experimental steps as well as the generation of waste and so on. Several aspects of the more desirable extraction results can be achieved.

### 2.1. Introduction to Green Extraction

The implementation of green chemistry in the extraction process can reduce or eliminate at source the use and release of chemical substances that are harmful to human health and the environment. “Green extraction” refers to the use of non-toxic solvents or alternative extraction methods in the extraction of the target [11], and usually the search for green reagents is one of the most important goals of green chemistry, extracting bioactive compounds from natural sources to replace the hazardous organic solvents currently in use.

The need for green extraction is based on the discovery and design of extraction processes that guarantee the optimal utilization of raw materials and solvents. Furthermore, one of the priorities set forth by the European Union in its environmental policy and legislation is to reduce the use of toxic solvents and volatile organic compounds, as most of these solvents are flammable, volatile and often toxic [12]. Consequently, a growing field of green extraction research is dedicated to the design of innovative, environmentally friendly and tunable solvents to meet technological and economic needs.

### 2.2. Application of Green Extraction Technology

At present, China has increasing attention toward green chemistry, and the development of green chemistry will also be a new direction of development. Solid-phase and liquid-phase microextraction techniques are also constantly developing and being applied. With continuous progress in the level of science, solid-phase microextraction and liquid-phase microextraction techniques are also becoming more and more diversified. The literature has reported that there are a lot of extraction materials applied in the pre-processing of samples, such as new extraction materials involving graphene [13], carbon nanomaterials [14], magnetic adsorbent materials [15], metal–organic framework materials [16] in solid-phase extraction, the current research topic, and, now for liquid-phase extraction of green chemical materials, deep eutectic solvents [17], ionic liquids [18], supramolecular solvents, and natural cotton fibers [19]. In particular, the introduction of carbon nanomaterials and deep eutectic solvents (DES) has triggered a research boom (Figure 1).

Green technologies are highly safe and environmentally friendly. With the development of carbon nanomaterials and the improvement of adsorption capacity, carbon nanomaterials have been widely used in various fields such as adsorption, medicine, bioengineering, and materials engineering due to their unique physical and chemical properties. Carbon nanomaterials show high efficiency affinity for interfacial chemical reactions, which is the key to realize the highly selective adsorption of polyphenol compounds in edible oils [20].

Compared with other green solvents, DES not only has the good characteristics of ionic liquids, in addition, it is simple to prepare, highly safe, and also has the advantages of very good biodegradability, and can maintain a transparent state at room temperature. However, DES has a high viscosity at room temperature and is not easy to be recovered, so when selecting suitable DES for extraction, careful consideration must be given to the physical and chemical properties of the extracted substances, as well as the physicochemical properties of the hydrogen bond donor and hydrogen bond acceptor, and also to the screening of solvents and recycling costs and other factors. The rational selection and use of green solvents are of great significance for the realization of an efficient and environmentally friendly green and high-efficiency extraction of target compounds.

## 3. Application of Novel Materials in Solid-Phase Extraction

Over the years, continuous advances in the field of extraction technology have revolutionized many perspectives from production quality to environmental safety. The development of new solid-phase extraction materials and pretreatment techniques capable of efficient, rapid and enrichment separation is a very important research topic. Traditional extraction methods, such as solid-phase extraction [21], liquid–liquid extraction and liquid–liquid microextraction [22], have played a wide range of applications in the extraction and enrichment of polyphenolic compounds from edible oils [23,24].

Based on the premise of green chemistry, with a low risk to human health and low impact on environmental pollution as the principle of development, a variety of novel extraction materials are widely used in the extraction process. Nano-magnetic carbon materials can be effectively applied to the extraction, purification and enrichment of small-molecule polar and nonpolar compounds in oil and grease samples with complex matrices, which can shorten the pretreatment time, improve the sensitivity, reduce the organic reagent consumption, and improve selectivity, for which the literature has reported many sample pretreatment applications of extraction materials. In this section, we will focus on the research progress of a series of materials such as carbon nanotube materials, graphene, metal–organic framework materials, magnetic adsorbent materials, and so on, in the study of polyphenolic compounds in edible oils. These novel materials have unique physicochemical properties, are environmentally friendly, and have high specificity for polyphenol compounds, and thus can rapidly and efficiently enrich and extract polyphenol compounds from edible oil samples, and some of them can simultaneously extract and detect polyphenol compounds, realizing the rapid enrichment and detection of polyphenol compounds, and thus have a good potential to be applied in the pretreatment of various types of complex samples for detecting the pretreatment technology of polyphenolic compounds in edible oil provides a reference.

### 3.1. Carbon Nanotubes

Carbon Nanotubes (CNTs) are a new kind of one-dimensional nanomaterial [25]; CNTs are the typical materials of hollow structure, which are nanotubular materials formed by graphene molecules aligned closely and seamlessly with each other [26]. CNTs have the characteristics of high mechanical strength, low density, resistance to acidity and alkalinity, and chemical stability [27], and the main production processes of CNTs are graphite arc [28], laser evaporation [29] and chemical vapor deposition [30]. CNTs were officially discovered by Japanese scientists Lijima et al. [31] in 1991. CNTs can be categorized into single-walled carbon nanotubes (SWCNTs) and multi-walled carbon nanotubes (MWCNTs) according to the number of layers of carbon atoms in their structure [32]. 

CNTs have great potential to be explored and applied in magnetic solid-phase extraction (MSPE) techniques due to their unique structure and excellent physicochemical properties [33]. MSPE is applied through the use of powdered magnetic materials or magnetizable adsorbents, which are not only homogeneously dispersed, but also reversibly agglomerated, to facilitate the extraction of polyphenolic compounds by the strength of the applied and removed external magnetic field, as well as the contact area between the CNTs’ magnetic adsorbent and the cooking oil contact area between the sample to facilitate the extraction of polyphenolic compounds.

In a kind of hydrophilic multi-walled carbon nanotube (HMWCNT) [34], in which polyphenolic compounds (trans-resveratrol) in edible oils can be successfully enriched and determined using MSPE, it was shown that HMWCNTs have strong adsorption properties and are excellent candidates for MSPE adsorbents. Rao et al. [35] prepared a new kind of magnetic carboxylated multi-walled carbon nanotube (c-MWCNT-MNP) for the determination of 23 polyphenolic compounds in sesame oil by magnetic solid-phase extraction–liquid chromatography–tandem mass spectrometry. When mentioning the main parameters affecting the extraction efficiency, which include the type and volume of desorption solvent, extraction and desorption time, and washing solution and adsorbent dosage, this suggests that MWCNTs have a promising application in the routine analysis of phenolic compounds in edible oil sample prospects. Zhao et al. [36] established a method based on the alendronate sodium grafted mesoporous magnetic nanoparticle (Fe_3_O_4_@ANDS) with high-performance liquid chromatography–ultraviolet-coupled detection, homemade magnetic composite HMWCNTs and Fe_3_O_4_ magnetite nanoparticles, which were assembled together as MSPE adsorbents, which can be used for the extraction and separation of trans-resveratrol from edible oils in a simple and efficient way. Ahmed et al. [37] used multi-walled carbon nanotubes as solid phase adsorbents to extract phenolic acids, flavonoids, and phenolics from 12 different samples with total contents ranging from 363 to 2658, 261 to 1646, and 224 to 1355 μg/100 g, respectively, and a total of 25 compounds were analyzed, which included 10 phenolic acids, 9 flavonoids, and 6 phenolics. Meanwhile, Pierantonio et al. [38] investigated the relative relationship between the amount of adsorbed biophenols and the number of carbon nanotubes used and the contact time, and it was necessary to reduce the number of carbon nanotubes used in order to obtain purer biophenol adsorbates, but an adequate number of carbon nanotubes ensured the selectivity of the biophenol adsorption in the presence of an excessive number of carbon nanotubes. This demonstrates not only the efficient extraction efficiency of multi-walled carbon nanotubes for phenolic compounds, but also the strong adsorption capacity for polyphenolic compounds. CNTs have been widely used in separation science as a novel material for sample extraction, and MWCNTs have great potential and offer a sizable alternative to perfect the preparation of magnetic adsorbent materials [39], while the method is applicable not only to edible oils but also to other fat matrices and analytes with different planar chemical structures.

### 3.2. Graphene

Graphene (G), the basic unit that makes up carbon nanomaterials, has been emphasized in various disciplines since its discovery in 2004. It is a novel sp² hybridized composite material with a two-dimensional lattice-like structure consisting of a single layer of tightly stacked carbon atoms, which is acid- and alkali-resistant, heat-resistant, electrically conductive, chemically stable, hydrophobic, and easily functionalized [40]. Compared with other materials, graphene has higher hardness and strength, with a tensile strength of up to 130 GPa and an elasticity coefficient of up to 1.1 TPa, which is more than tens of times that of steel. As shown in Figure 2, graphene and its derivatives are mainly classified into two categories: graphene oxide (GO) and reduced graphene oxide (rGO) [41], and GO not only has a super-large specific surface area, but also carries reactive groups, such as carboxyl and carbonyl, with good adsorption properties.

Graphene is a large aromatic ring lamellar structure that can be utilized to have strong interactions between the aromatic ring structures of compound molecules; therefore, graphene as an adsorbent is the best to be used for the extraction of small-molecule compounds containing aromatic rings. It has been found that graphene, GO and rGO are adsorbents for microsolid-phase extraction. In addition to better purification in the pretreatment solution, graphene can be used in the treatment of solid samples for organic compounds and trace metal ions for the adsorption of food, biological and environmental samples, etc., where it has a good adsorption effect [42]. Sahin et al. [43] used GO synthesized by Hummer’s method as an adsorbent for hydroxytyrosol and the adsorption of this bioactive compound from aqueous medium onto GO was >85% under optimum conditions. The pH of the adsorption medium was found to be a very important parameter affecting the recovery of hydroxytyrosol, increasing the pH from 3 to 9, and increasing the adsorbate content of GO from 0.55 mg to 89.46 mg.

Graphene nanoparticles (GNP) have all the superior properties of graphene; in addition, GNP is more stable than monolayer graphene. Sahin et al. [44] devised a method for the recovery of olive picosides and hydroxytyrosol from olive leaves and olive oil was investigated, for which a nanocomposite of zirconium-based metal–organic skeleton (UiO-66) and GNP was successfully prepared for the exploration of different characterization techniques for the morphology and structural properties of UiO-66 and GNP/UiO-66 nanocomposites. The results showed that this adsorbent has a good affinity for olive bittersweet and hydroxytyrosol. Metal oxides are effective catalysts and graphene is a good support platform for nanoparticles, so combining them together should result in better nanocomposite catalysts. Apinya et al. [45] prepared a novel zirconium dioxide (ZrO), cobalt oxide (CoO), and rGO nanocomposite catalyst material by means of fluorine-doped tin oxide electrode. This material was made into an electrochemical sensor, and due to the synergistic effect between metal oxide nanoparticles and graphene, it has been shown that the method efficiently and selectively will be used for the determination of gallic acid, caffeic acid and protocatechuic acid in the sample.

Graphene quantum dots (GQDs) [46] are emerging light-emitting carbon nanomaterials that are not only strongly chemically inert, but also have excellent photostability [47], and these properties endow GQDs with a variety of potential uses in the presence of photovoltaic devices, bioimaging instruments, and biosensors, among others [48]. Martínez et al. [49] utilized graphene to develop a novel quantum dot optical nanosensor, obtained via the pyrolysis of citric acid, for the determination of common model analytes of polyphenolic compounds such as gallic acid and olive bitters in olive oil, as well as the concentration of polyphenolic compounds in real samples of olive oil. In addition, graphene has a wide range of applications in the fields of polymer material science, energy dynamics and biomolecule drug delivery; thus, graphene has been hailed as an epoch-making new material. 

### 3.3. Metal–Organic Frameworks

Metal–organic frameworks [50] (MOFs), also known as porous copolymers (PCPs), are periodically hybridized crystal structures formed by metal cations and organic ligands through coordination [51]. Compared with traditional materials, MOF materials have a very desirable stability for the extraction and recovery of polyphenolic compounds due to their large porosity, large specific surface area, and diverse structures and functions, which are widely used in gas adsorption and separation technology, sensor technology, drug release technology, and many catalytic reactions [52]. Meanwhile, metal–organic frameworks are used in polyphenolic compounds, furans, ketones, wheat oil, carboxylic acids and sugars for extraction and enrichment. It was shown that a highly selective adsorbent for polyphenolic compounds was selected from four metal–organic skeleton materials; the four materials were MIL-53(Al), MIL-53(Cr), MIL-47(V), and Basolite A100, among which the MIL-53(Al) material had the best adsorption capacity, with optimal adsorption capacity and good selectivity [53].

Magnetic metal–organic framework (MMOF) materials combine different MOF materials with various magnetic nanoparticles, adding magnetic properties to the advantages of MOF materials in terms of material selectivity, biocompatibility, ease of manipulation, and reproducibility. In combination with solid-phase extraction (SPE) technology, MMOF materials have been widely used in the pretreatment of complex matrices, such as pesticide and veterinary drug residues, toxins, food additives, and heavy-metal ions in foods. In the extraction of polyphenolic compounds and enrichment and separation of trace pollutants in edible oils, aflatoxin in vegetable oils was extracted by employing copper-based metal–organic framework (Cu-BTC MOF)-derivatized porous materials as adsorbents, and the experiments showed that the C-Cu-BTC MOF materials not only removed more than 90% of AFB in edible oils within 30 min, but also had a lower level of AFB in the treated oils. Oil still has a low cytotoxicity, while the adsorption process has little effect on the quality of edible oil [54]. Therefore, MOF materials are characterized by high efficiency, safety, practicality and economy, and can be used as new potential adsorbents.

Covalent organic frameworks (COFs) are an emerging class of porous crystalline pharmaceutical macromolecular materials that have been developed in recent years [55]. COFs have been used in the analysis of food contaminants due to their advantages of low density, large surface area, high safety, and ease of functional remediation [56]. The main applications of COF adsorbents in the analysis of food contaminants include the extraction of complex samples and the extraction of organic contaminants and heavy metal ions from food contaminants. Kang et al. [57] combined nano-optical sensors with near-infrared spectroscopy for the rapid detection and quantification of polyphenolic compounds and investigated the potential of chemoselective colorant-based nano-optical sensors for detecting the dynamic changes in aroma constituents during the fermentation of extracts, which demonstrated an improvement in the efficiency of polyphenolic compound detection at an ultrasound frequency of 28 kHz, a processing time of 24 min, and an ultrasound power density of 40 W/L. In magnetic porous organic frameworks (MPOFs) materials, due to the presence of pores, the physical and chemical properties of porous materials have been greatly enhanced, with the remarkable features of diverse synthesis methods, high physicochemical stability, strong magnetic responsiveness, easy separation, and excellent application performance in the extraction and enrichment of the target analytes for organic compounds (veterinary drugs, pesticides, endocrine disrupters, and other organic pollutants), heavy-metal ions and biomolecules, etc., which have a wide range of applications [58].

In recent years, although some breakthroughs have been made in food safety and detection with novel magnetic nanomaterials based on carbon nanotubes, graphene, and metal–organic frameworks, these materials are mainly used to achieve selective adsorption and efficient enrichment of trace analytes through hydrophobic/hydrophilic interactions, and hydrogen bonding. The commonly used assistive techniques include microwave-assisted extraction [59], ultrasound-assisted extraction [60], high-voltage pulsed electric field extraction [61], heated reflux extraction [62], and supercritical fluid extraction [63]. At present, the auxiliary technology has been developed from the laboratory to the industrial application; magnetic adsorbents will simplify the process of specimen pretreatment in the extraction of polyphenol compounds in edible oils, improve the recycling rate of the test, and have a certain significance for the further development of pretreatment materials.

## 4. Application of Green Chemical Materials in Liquid-Phase Extraction

Liquid-phase microextraction (LPME) [64] is a small-scale environmentally friendly pretreatment technology that extracts, purifies, and concentrates the target in a single unit, which has the advantages of good selectivity, simple operation, low labor intensity, less organic solvents required, and low pollution, and overcomes many shortcomings existing in the traditional liquid–liquid extraction (LLE) technology [65], and uses only micro-upgrading, or even nano-upgrading, of the organic solvents used in the extraction process. Upgrade is not only a green analytical technology, but also a new analytical technology in line with the development of microextraction in modern analytical science, but also a new sample separation and enrichment technology that has attracted much attention.

“Green solvents” are a class of liquid phases used for the recovery of many biologically active compounds, which are used in the food, agriculture and biotechnology industries. Although water is an excellent green solvent, it is not effective for the extraction of hydrophobic substances. Currently, mainly room temperature ionic liquids, deep eutectic solvents, supercritical carbon dioxide, supramolecular solvents and custom synthesized green solvents have been used in bioactive extraction [66]. In this section, we will focus on various green solvents and their research progress in the extraction of polyphenolic compounds, and future research trends may focus on further improving the properties of green solvents and developing more efficient liquid-phase extraction materials. 

### 4.1. Deep Eutectic Solvent

Deep eutectic solvents (DES) are the new generation of solvents made from renewable energy sources, unlike natural deep eutectic solvents (NADES), which are solvents made from natural phytometabolites. DES are obtained by mixing and stirring the hydrogen bond donor (HBD) and hydrogen bond acceptor (HBA) at a specific temperature for a period of time with a certain molar ratio [67]. DES is an efficient, safe, low-cost and biodegradable green solvent with physicochemical properties such as a low melting point, higher relative density than water, low volatility at room temperature, nonflammability, and high viscosity. Not only that, DES can also be used in conjunction with magnetic nanomaterials to adsorb target analytes through hydrogen bonding, π-π forces and electrostatic force specificity. Currently, DES have shown incredible capabilities in the extraction and separation of biologically active ingredients, residual drugs, metal ions and organics from complex matrices.

In the preparation of DES, based on inexpensive or readily available ingredients, usually consisting of two or more substances, which include nontoxic quaternary ammonium hydrogen bond acceptors, such as choline chloride and betaine [17], and naturally derived uncharged hydrogen bond donors, such as amides, sugars, alcohols, and carboxylic acids [68] (as shown in Figure 3), DES is produced when the hydrogen bond donor and hydrogen bond acceptor are mixed in a certain molar ratio, and the DES is produced when the mixture has a eutectic point with a low melting point. Relative to ionic liquids, its physical and chemical properties are similar to those of ordinary ionic liquids, such as viscosity, electrical conductivity, density, surface tension, refractive index, and chemical inertness, etc. At the same time, DES has the advantages of being inexpensive, natural, of high purity, and safe for the environment, so DES is a green solvent that has the promise of being able to replace ionic liquids [69]. The main disadvantage of DES is that it has a high viscosity and density, which requires desorption of PCs from the resin during the extraction process with mobility, in which a large amount of volatile solvents are consumed [70].

DES was first reported by Abbott [71] in 2003, who successfully synthesized a eutectic mixture of choline chloride and urea at a molar ratio of 1:2, which was analyzed and showed physicochemical properties close to ionic liquids with good biodegradability and biocompatibility. Paradiso et al. [72] used glucose and lactose as the raw materials for the preparation of DES, and the 65 extra virgin olive oils. The DES extracts of phenolic compounds in the samples were subjected to spectrophotometric characterization and total phenol content, and methanol–water solution extraction was used as a control combined with analysis, and the spectrophotometric method was used to screen the total phenol content of vegetable oils. And DES has also been applied in rotating disk sorptive extraction (RDSE), which is considered as an effective sample preparation method, and a typical rotating disk is a polymer disk equipped with an adsorbent phase and a stirring magnetic bar. Andrey et al. [73] proposed a rotating disk sorptive extraction based on the mechanism of DES formation and used it for the extraction and separation of polyphenolic compounds in olive oil samples, and this method can significantly reduce the adsorption time and elution time, while various applications of DES and NADES for the extraction of polyphenolic compounds in edible oils are mentioned in Table 1, which shows that this material is a solvent with high solubilizing capacity and can be used as a green solvent for the extraction of a wide range of bioactive molecules. Thus, DES and NADES can be a viable method for the green extraction of polyphenolic compounds from edible oils. However, there is a lack of research on the theoretical mechanism of DES, microstructure analysis between substances, and extraction and separation mechanism. The physicochemical properties of substances are the basis of their application; therefore, emphasis should be placed on exploring the potential theoretical mechanisms of deep eutectic solvents and developing new deep eutectic solvents with special functions.

### 4.2. Ionic Liquids

Ionic liquids (ILs) [18] are new green solvents that replace conventional organic solvents, are low-temperature molten salts with virtually no vapor pressure, and are potential alternatives to conventional volatile organic solvents used for extraction. ILs are characterized by their chemical stability, high solubility, structural design, and high electrical conductivity. The physical and chemical properties of ILs are dependent on the nature and size of the cations and anions, which gives unique properties and, therefore, these reagents are also known as “designer extractants” [81].

ILs are mainly composed of organic cations (organic nitrogen-containing heterocycles) and a variety of anions, which are generally synthesized by direct synthesis and ion-exchange, etc., and are liquid at room temperature. As shown in Table 2, there are 2 main types of ILs [82]: the first type is divided into imidazole cations (e.g., 1-butyl 3-methylimidazolium bromide), pyridine cations (e.g., N-butylpyridinium bromide), quaternary phosphine cations (e.g., tetrabutylphosphonium bromide), and quaternary ammonium cations (e.g., tetraethylammonium bromide), according to the structures of their cations; and the second type is divided into two types according to the structure of their anions: one type is halogenated salts and non-halogenated salts. Classical IL anions include bisimine (NTF2), trifluoromethyl sulfate (TFO), dicyanamide (N(CN)2), tetrafluoroborate (BF4), or hexafluorophosphate (PF6). And simple anions have also been found in less stable ILs or room-temperature nonliquids, including chloride, bromide, iodide, nitrate, perchlorate, formate and acetate [83]. Although there are many similarities between DES and ILs, there are significant differences in their structure, appearance, spectra and synthesis, where ILs are stabilized by the electrostatic attraction of anions and cations, whereas deep eutectic solvents are formed by hydrogen bonding and van der Waals forces between components [84]. In fact, ILs can be stabilized in most of the solvents, so the thermal and chemical stability of ILs is better than that of deep eutectic solvents.

Walden et al. already reported an ethylamine nitrate ([EtNH_3_] [NO_3_]) with a melting point of 12 °C as early as 1914, which was the first organic salt to present a liquid state at room temperature and is also considered to be the earliest IL. Quaternary ammonium salt ILs can be referred to as room-temperature ionic liquids or room-temperature molten salts (RTILs) when they present a liquid state at or near room temperature [85]. Not only that, ILs can be easily immobilized on solid carriers by physical or chemical means (e.g., silica gel [86], polymers [87], and graphene oxide [88] as carriers), thus obtaining novel extraction materials with internally embedded ionic liquids or decorated on surfaces. In addition, ILs can be converted into ionic liquid immobilized composites by impregnation and grafting. The resulting composites are characterized by high enrichment efficiency, high adsorption capacity, good stability, and high extraction selectivity, as well as multiple ionic liquid recognition sites and high utilization [89]. In recent years, ILs have attracted the attention of scientists due to their unique physical and chemical properties and intermolecular interactions, and they have been widely used as solid-phase extraction adsorbent materials for the separation of organic small molecules, as well as having a wider range of applications in the food field [90].

The application of ILs has been reported in the extraction and isolation of plant constituents such as phenolic compounds, flavonoids, glycosides, alkaloids, aromatic compounds, and essential oils. Zhu et al. [91] used magnetic ionic liquids as microextraction solvents to prepare a new method for the extraction of two endocrine-disrupting chemicals, bisphenol A and bisphenol A, and 4-nonylphenol in vegetable oils, which was extracted by dispersive liquid–liquid microextraction, and determined by ultra-high-performance liquid chromatography–tandem mass spectrometry for the determination of two polyphenols in vegetable oils. Zhang et al. [92] prepared a novel three-dimensional ionic liquid-functionalized magnetic graphene oxide nanocomposite (3D-IL@mGO), which was used as an effective adsorbent for the magnetically dispersive solid-phase extraction of 16 polycyclic aromatic hydrocarbons (PAHs) in vegetable oils and analyzed by gas chromatography–mass spectrometry (GC-MS). The properties of the 3D-IL@mGO material were also characterized. It was demonstrated that the 3D-IL@mGO material had been functionalized by ionic liquids and exhibited high adsorption performance and good stability for the extraction of PAHs in vegetable oil samples.

### 4.3. Supercritical Fluid Extraction

The supercritical fluid extraction (SCFE) technique is an effective separation research technique, which has been rapidly developed and widely used in recent years [63]. The three-phase point of a substance is the coexistence of three forms, i.e., solid, liquid, and gas, at a specific temperature and pressure. The critical point is when the phase difference between solid, liquid and gas decreases as the temperature and pressure increase. At this critical point, the substance has the solubility and density of a liquid and behaves like a gas [93]. SCFE extracts biologically active substances from plants at atmospheric temperatures and prevents their thermal denaturation, which has attracted a lot of attention due to its simple design and construction. The most common solvents are hexane, pentane, toluene, nitrous oxide and sulfur hexafluoride, However, carbon dioxide (CO_2_) is the most widely used solvent because CO_2_ is the easiest and least costly solvent to remove, the critical point of CO_2_ exists at a critical temperature of 31 °C and a critical pressure of 74 bar, and, due to its supercritical properties, it is considered to be a promising solvent [94]. Subcritical fluid extraction (SFE) utilizes the two-phase change in the extraction solution at different temperatures and pressures to accomplish the process of edible oil extraction and extraction solvent removal. In the subcritical state, water, petroleum ether, n-butane, methanol, and ethanol are used to extract vegetable oils. Among them, subcritical water, with a critical temperature of 100–374 °C and a critical pressure of 1–22.1 MPa, has become the most commonly used extractant because of its non-toxicity, low cost and high diffusion rate. Compared with traditional extraction techniques, SFE can shorten the extraction time, improve the quality of crude oil, reduce energy and solvent consumption, and increase the yield [95].

Supercritical CO_2_ (SCCO) is highly stable, non-corrosive, non-flammable, inexpensive, colorless and odorless, making it one of the ideal choices for isolation and purification in the food industry. SCCO’s low polarity makes it unsuitable for extracting polar molecules, but it can be improved by adding polar-entraining agents (e.g., water, ethanol, etc.), and it is the “ideal” green extraction solvent, and it has the advantages of non-toxicity, non-combustibility, and low cost, etc. It is widely used in the extraction of proteins, amino acids, and polyphenolic compounds in oilseed food materials, as well as fatty acids and polyphenolic compounds (Table 3). Fang et al. [96] explored the effect of different extraction methods (HPE, AEE, SE and SCCO) on the content of β-sitosterol, squalene and tocopherols in vegetable oils, and the results showed that the highest content of β-sitosterol, squalene and tocopherols was found in vegetable oils extracted using SCFE-CO_2_. Mohamed et al. [97] used the n-hexane extraction technique and the emerging SCCO extraction technique. The extraction efficiency of polyphenolic compounds from seed oil extraction red prickly pear was compared, and the polyphenolic compounds were identified and quantified by gas chromatography and ultra-high-performance liquid chromatography–high-resolution mass spectrometry, respectively. The efficiency of SCCO extraction as a green process was demonstrated by the ability to identify 45 polyphenolic compounds and obtain a higher polyphenolic content using the SCCO extraction technique, although the same oil yield was obtained when compared to the n-hexane extraction technique. However, there are some drawbacks of SCCO while considering important parameters such as temperature, pressure, sample size, solvent addition, and flow rate control. Balancing each other plays a crucial role in designing highly sensitive processes with a high number of operating conditions [98]. Negative aspects aside, supercritical fluid extraction has shown positive signs when compared to other conventional techniques for the extraction of polyphenolic compounds.

The development of SCFE technology overcomes the shortcomings of conventional extraction processes such as the toxicity of organic solvents, high energy consumption, and low extraction efficiency, etc. SCFE technology has a broad application prospect, and it needs to be studied in depth to find out its types, operating conditions, and technological costs, etc., and be improved with a view of achieving commercialization and use.

### 4.4. Supramolecular Solvents

Supramolecular solvents (SUPRAS) are essentially colloids, which are discontinuous liquids composed of aggregates of amphiphilic compounds, and nanostructured liquids generated by two principles: assembly alone and coalescence phenomena [106]. Solo assembly is designed based on the structure of the desired molecule or its surroundings, while coalescence is based on the separation of the colloid into two fluid components. The equilibrium phase and the condensed phase, respectively, with the latter having a higher concentration of colloids than the former. In recent years, considerable progress has been made in the application of SUPRAS through the extraction of biologically active compounds. SUPRAS not only has unique physicochemical properties, but it is also highly amphiphilic and at the same time adjusts the hydrophobicity of amphiphilic substances, among many other advantages.

To date, the most commonly used amphiphilic reagents for the extraction of biologically active compounds are nonionic surfactants. In order to extract polyphenolic compounds more efficiently, SUPRAS tend to polymerize with amphiphilic groups whose polymerization involves both nonpolar and polar reactions. It has been found that amphiphilic molecules carrying the same charge react more strongly with opposite charges than with the same charge. Elevation of the temperature of the hydrocolloid solution or addition of water to the organocolloid solution can induce the formation of SUPRAS [107].

The composition and nanostructure of SUPRAS will greatly influence the amount and distribution of extracted bioactive compounds, and the extraction efficiency of bioactive compounds is superior to that of conventional organic solvents. Keddar et al. [108] investigated the extraction of total carotenoids and polyphenolic compounds by using different SUPRAS compositions as lipophilic and hydrophilic antioxidants, respectively. It was demonstrated that lipophilic and hydrophilic antioxidants interact with each other in vivo through different mechanisms to maximize the total phenolic content extracted by using SUPRAS as extracts while optimizing the composition and extraction parameters of SUPRAS. Deep eutectic supramolecular polymers (DESPs) [109] are a new type of green extraction solvents, which are derivatives of DES. DESP not only possesses the properties of DES as a multifunctional medium, but is also a medium for the efficient extraction and stable storage of biologically active components. Shi et al. [110] prepared a DESP, which was composed of a supramolecular polymer β-cyclodextrin (β-CD) as the HBA and lactic acid (LA) as the HBD, in a ratio of 1:1 (*w*/*w*%), while ultrasound-assisted DESP extraction was used for the best extraction efficiency of polyphenolic compounds.

Thus, SUPRAS can play an important role in the extraction of bioactives and the nature of SUPRAS allows them to replace conventional organic solvents in the extraction process.

## 5. Conclusions and Perspectives

The technology of the extraction and separation of polyphenolic compounds from edible oils is developing rapidly, in which the more environmentally friendly green extraction methods provide a better alternative to the traditional methods. In this paper, solid-phase adsorbent materials and liquid-phase extraction materials are the main research directions, and the advantages of carbon nanotubes, graphene, metal–organic frameworks, deep eutectic solvents, ionic liquids, supercritical fluids, and supramolecular solvents for extracting polyphenol compounds from edible oils are described in detail.

Future research trends may further improve the properties of green solvents and the development of more efficient solid-phase adsorption materials and liquid-phase extraction materials. The combined use of multiple extraction methods often has the advantage of overcoming the limitations of a particular method, and considering the composition and sensitivity of bioactive molecules, detailed studies are needed to understand their synergistic mechanisms and application applicability. In addition, sample preparation is a critical step for the efficient extraction of target compounds prior to their extraction and separation. In conjunction with the current status of green extraction techniques for polyphenolic compounds in edible oils, other potential application possibilities can be explored, such as pharmaceutical formulations, food additives and nutraceuticals. Combined with the development of green technology, this will provide more options and development space for the extraction process of polyphenols in edible oils.

## Figures and Tables

**Figure 1 molecules-28-08150-f001:**
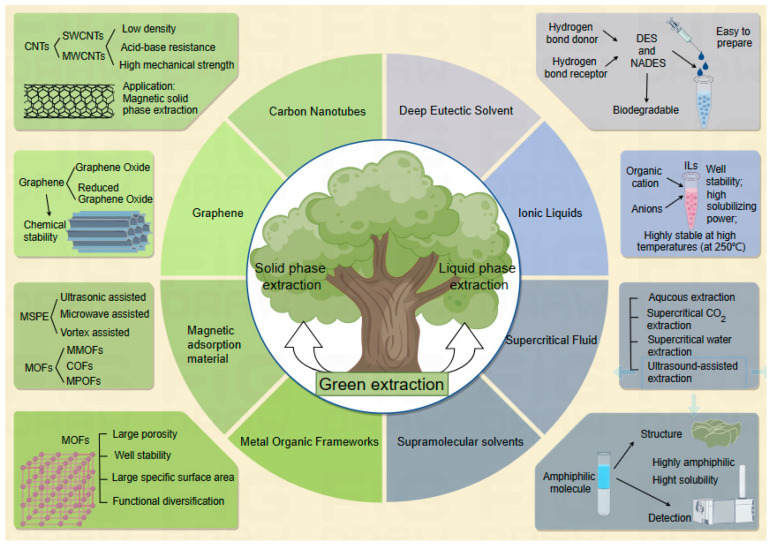
Different green extraction techniques.

**Figure 2 molecules-28-08150-f002:**
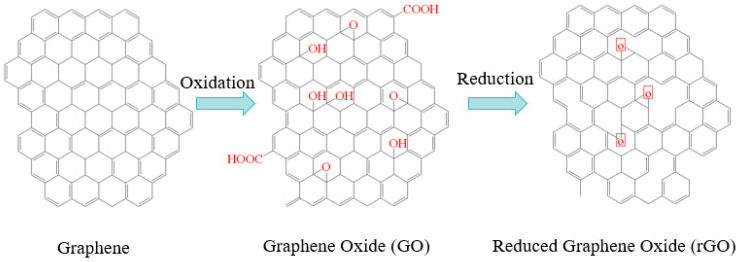
Structure of graphene.

**Figure 3 molecules-28-08150-f003:**
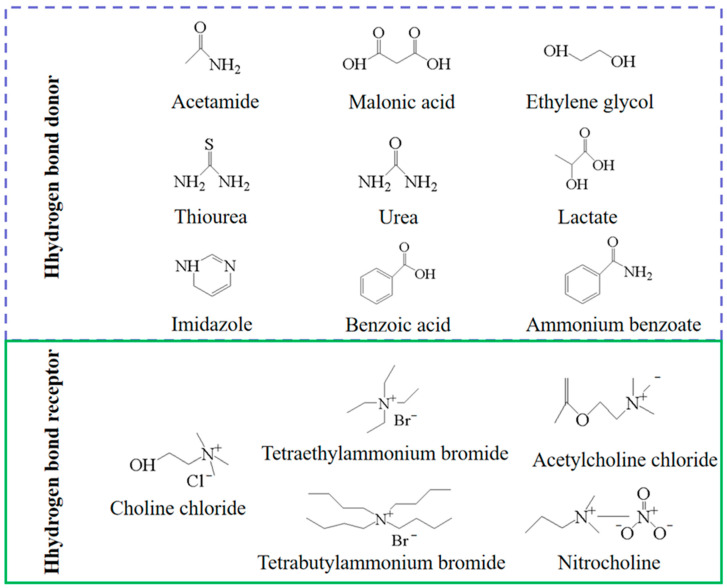
Common hydrogen bond donors and hydrogen bond acceptor.

**Table 1 molecules-28-08150-t001:** Application of DES and NADES.

Number	NADES	Extraction Method	Source	Target Compound	Detection Method	Reference
1	Lactic acid:Glucose:Water 6:1:6	Liquid–liquid extraction	Olive oil	Total phenol content	Direct Spectrophotometric Detection	[72]
2	Apply choline chloride to the surface of the rotating disc	Rotary disk adsorption extraction	Olive oil	Gallic acid, protocatechuic acid, tyrosol, vanillic acid, p-coumarinic acid, syringaldehyde, thymol	HPLC-FLD	[73]
3	Citric acid:Fructose 1:1 (19% water)	Liquid–liquid extraction	Olive oil	3,4 dihydroxyphenyl glycol, hydroxytyrosol glucoside, hydroxytyrosol, tyrosol glucoside (salidroside), verbascoside, dialdehydic form of decarboxymethyl oleuropein aglycon (3,4-DHPEA-EDA), caffeoil-6′-secologanoside, comselogoside	HPLC-ESI-MS	[74]
4	Choline chloride:Xylitol:Water 2:1:3	Liquid–liquid extraction	Olive oil	Hydroxytyrosol, Oleacin, oleocanthal, oleuropein, aglycone, ligstroside, aglycone, 1-acetoxypinoresinol, tyrosol, luteolin, apigenin, pinoresinol	HPLC-DAD	[75]
5	Betaine:Glycerol 1:2 (30% water)	Liquid–liquid extraction	Olive oil	Hydroxytyrosol, tyrosol, dialdehydic form of oleuropein aglycon, oleuropein aglycon isomer, lygstroside aglycon	HPLC-DAD/ESI-MS	[76]
6	Lactic acid:Glucose:Water 3:1:3	Liquid–liquid extraction	Olive oil	Hydroxytyrosol, tyrosol derivatives	Direct Spectrophotometric Detection	[77]
7	Lactic acid:Glucose:Water 6:1:6	Liquid–liquid extraction	Olive oil	Benzoic acid derivatives (hydroxybenzoic acid, protocathecuic acid, vanillic acid), cinnamic acid derivatives (p-coumaric acid, caffeic acid), phenyl-ethyl alcohols (tyrosol), flavonoids (apigenin), lignans (pinoresinol)	Direct Spectrophotometric Detection	[78]
8	Choline chloride:Ethylene glycol 1:2, Choline chloride:Glycerol 1:2	Ultrasonic-assisted liquid–liquid microextraction method	Olive oil; Sesame oil; Cinnamon oil; Almond oil;	Ferulic, caffeic, cinnamic	HPLC-UV	[79]
9	Choline chloride:xylitol Choline chloride/1,2-propanediol	Liquid–liquid microextraction	Olive oil	Oleacein, Oleocanthal,	UPLC-DAD/MS	[80]

**Table 2 molecules-28-08150-t002:** Classification of constituent ionic liquids.

Cation	Anion	Cation + Anion	Example
Pyridine cation	Halogenation Non-Halogenated	Pyrazole Halogenated Pyrazole Non-Halogenated	N-Butylpyrazole bromide N-Butylpyrazole tetrafluoroborate
Imidazole cation	Halogenation Non-halogenated	Imidazole halogenated Imidazole Non-Halogenated	1-Butyl-3-methylpyrazole bromide 1-Butyl-3-methyl tetrafluoroborate
Quaternary ammonium cation	Halogenation Non-halogenated	Quaternary ammonium halogenated Quaternary ammonium non-halogenated	Tetraethylammonium bromide Tetrabutyl tetrafluoroborate
Quaternary phosphoric cation	Halogenation Non-halogenated	Quaternary phosphine halogenated Quaternary phosphine non-halogenated	Tetrabutylphosphonium bromide Tetrabutylphosphonium tetrafluoroborate

**Table 3 molecules-28-08150-t003:** Applications of supercritical fluids extraction.

Number	Supercritical Fluid Extraction	Source	Target Compound	Optimum Extraction Condition	Reference
1	Supercritical fluid extractio, subcritical fluid extraction,	Camellia oleifera oil	squalene, tea polyphenols, tocopherol, phytosterol.	extraction temperature of 45 °C	[99]
2	Supercritical CO_2_	Moringa oleifera seed oi	unsaturated fatty acids, including oleic acid, octadecanoic acid, palmitic acid, stearic acid, eicosanoic acid.	extraction temperature of 45 °C; extraction time of 2.5 h; extraction pressure of 50 MPa; CO_2_ flow rate of 240 L/h; resulting in a maximum yield of 38.54%.	[100]
3	Supercritical CO_2_	Yellow horn seed oil	oleic 33.40%, linoleic acids 44.81%, nervonic acid 4.06%.	extraction temperature of 50 °C; extraction time of 2 h; extraction pressure of 40 MPa; resulting in a maximum yield of 31.61%.	[101]
4	Supercritical CO_2_ and EtOH as co-Solvent	Sunflower edible oil	total polyphenol content, total flavonoids content,	extraction pressure 21 MPa; extraction temperature of 60 °C; co-solvent contribution 15% *w*/*w* EtOH	[102]
5	Supercritical CO_2_	Berberis dasystachya Maxim Seed Oil	unsaturated fatty acids (85.62%) and polyunsaturated fatty acids (57.90%), aldehydes, esters.	extraction pressure 25.00 Mpa; extraction temperature of 59.03 °C; CO_2_ flow rate of 2.25 SL/min; maximum yield of 12.54 ± 0.56 g/100 g	[103]
6	Hexane extraction (HE), aqueous enzymatic extraction (AEE), supercritical fluid extraction (SFE).	Xanthoceras sorbifolia Bunge Kernel Oil	monounsaturated fatty acids (49.31–50.38%), oleic acid (30.73–30.98%), nervonic acid (2.73–3.09%), HE: resulted in the highest oil yield (98.04%), tocopherol content (530.15 mg/kg), sterol content (2104.07 mg/kg).	HE: extraction temperature of 50 °C; extraction time of 4 h. SFE: CO_2_ flow rate of 18 L/h; extraction pressure of 28 MPa; extraction temperature of 42 °C.	[104]
7	Cold pressing (CP), microwave pretreatment pressing (MP), supercritical fluid extraction (SFE)	Pumpkin Seed Oil	total polyphenol content, squalene, tocopherols, phytosterols.	SFE > MP > CP	[105]

## Data Availability

Data are contained within the article.

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
