# Peer review of "Recent Progress on Green New Phase Extraction and Preparation of Polyphenols in Edible Oil"

_molecules, 2023, doi:10.3390/molecules28248150_

Round 1
Reviewer 1 Report
Comments and Suggestions for Authors
The manuscript offers a comprehensive and informative overview of green extraction techniques, particularly focusing on the extraction of polyphenols from edible oils. It effectively highlights the relevance of green chemistry and explores novel materials in extraction processes. However, there are areas where the manuscript could be improved to enhance clarity, depth, and overall impact.
Some comments need to be considered before the paper is accepted for publication:
Abstract (Lines 48-59)
The abstract should more clearly state the main findings or conclusions of the study. Additionally, a brief mention of the research gap or the novelty of the study would enhance its impact.
- **Recommendation**: Revise the abstract to include the main findings and highlight the study's novelty.
Introduction (Lines 80-91)
The introduction's discussion on the background could be more concise. It is essential to explicitly state the research objectives or questions, which are currently only implied.
- Recommendation: Refine the introduction to concisely state the background and explicitly outline research objectives or questions.
Research Progress on Green Extraction Techniques (Lines 222-247)
This section would benefit from a clearer structure, perhaps with subheadings for each extraction technique. Comparative insights or a critical evaluation of these techniques in terms of efficiency or applicability are needed.
- Recommendation: Reorganize the section with subheadings and include comparative insights or evaluations.
Progress of Research on Novel Materials in Solid-Phase Extraction (Lines 315-332)
The section lacks specific examples or case studies demonstrating the effectiveness of novel materials in real-world applications. A discussion on limitations or challenges associated with these materials is also necessary for a balanced view.
- Recommendation: Incorporate specific examples or case studies and discuss any limitations or challenges.
Research Progress on Green Chemical Materials in Liquid Phase Extraction (Lines 676-687)
Like previous sections, this part needs a more detailed comparison of these methods with traditional techniques, including specific efficiency metrics.
- Recommendation: Enhance the section with detailed comparisons and efficiency metrics.
Figures and Tables
- Specific Comment on Figure 3: The figure presents common hydrogen bond donors and acceptors but omits more common examples like ChCl and Gly. Consider expanding the range of examples for completeness or explain why those have been cited rather than others.
Conclusion (Lines 1417-1431)
The conclusion should provide more specific future research directions or potential applications of these techniques in industry or academia.
- Recommendation: Expand the conclusion to include specific future research directions and potential applications.
References
- There is a need to assess the relevance and recency of references. Only two sources from 2022 are cited, which may not adequately reflect current research trends.
- Recommendation: Update the literature review to include more recent and relevant sources.
Author Response
Point-to-point responses
Thank you very much for giving us an opportunity to revise our manuscript. We have studied the comments. Any revisions have been marked in “Red” in revised manuscript. We appreciate for your warm work earnestly, and hope that the correction will meet with approval. Once again, thank you very much for your comments and suggestions. The point-to-point responses were listed as follows.
Reviewer #1
- Abstract (Lines 48-59):
The abstract should more clearly state the main findings or conclusions of the study. Additionally, a brief mention of the research gap or the novelty of the study would enhance its impact.
- **Recommendation**: Revise the abstract to include the main findings and highlight the study's novelty.
Thank you very much for your suggestion, the summary section (lines 16-26) has been reworked according to your suggestion.
- Introduction (Lines 80-91):
The introduction's discussion on the background could be more concise. It is essential to explicitly state the research objectives or questions, which are currently only implied.
- Recommendation: Refine the introduction to concisely state the background and explicitly outline research objectives or questions.
The introduction section (lines 28-81) has been reworked in response to your suggestions. The logic of the introduction has been reorganized to more concisely state the objectives of the study.
- Research Progress on Green Extraction Techniques (Lines 222-247):
This section would benefit from a clearer structure, perhaps with subheadings for each extraction technique. Comparative insights or a critical evaluation of these techniques in terms of efficiency or applicability are needed.
- Recommendation: Reorganize the section with subheadings and include comparative insights or evaluations.
Based on your suggestions have reworked Chapter 2, Advances in Research on Green Extraction Technology (lines 75-126), adding subheadings, 2.1. Introduction to green extraction and 2.2. Application of green extraction technology, and improving the content.
- Progress of Research on Novel Materials in Solid-Phase Extraction (Lines 315-332):
The section lacks specific examples or case studies demonstrating the effectiveness of novel materials in real-world applications. A discussion on limitations or challenges associated with these materials is also necessary for a balanced view.
- Recommendation: Incorporate specific examples or case studies and discuss any limitations or challenges.
Chapter 3, Application of Novel Materials in Solid Phase Extraction (lines 129-154), has been reworked in response to your suggestions to add a discussion of the limitations of these materials.
New case studies and summaries have been added to 3.1 Carbon nanotubes (lines 188-200); 3.2 Graphene (lines 226-236); and 3.3 Metal-organic frameworks (lines 285-291).
- Research Progress on Green Chemical Materials in Liquid Phase Extraction (Lines 676-687):
Like previous sections, this part needs a more detailed comparison of these methods with traditional techniques, including specific efficiency metrics.
- Recommendation: Enhance the section with detailed comparisons and efficiency metrics.
Based on your suggestions have reworked Chapter 4, Application of Green Chemical Materials in Liquid Phase Extraction, by adding the following sections:
- comparison of green extraction solvents (lines 302-310)
- potential mechanism of DES (lines 319-323) and outlook (lines 358-363)
- Suitability of Ionic Liquids for Extraction of Various Compounds in Edible Oils (Lines 409-411)
- table of supercritical fluids extraction Table 3. applications of supercritical fluids extraction (line 477).
- citation and summary of supramolecular solvents (lines 506-512).
- Figures and Tables:
- Specific Comment on Figure 3: The figure presents common hydrogen bond donors and acceptors but omits more common examples like ChCl and Gly. Consider expanding the range of examples for completeness or explain why those have been cited rather than others.
Hydrogen bond donors and hydrogen bond acceptors are not presented in abbreviated form in Figure 3, you mentioned ChCl as the abbreviated form of Choline chloride in Figure 3. Because there are many kinds of hydrogen bond donors and hydrogen bond acceptors, and most of the cases will be seen in some articles, instead of ignoring some uncommon and special hydrogen bond donors and hydrogen bond acceptors, I have summarized a variety of hydrogen bond donors and hydrogen bond acceptors.
- Conclusion (Lines 1417-1431):
The conclusion should provide more specific future research directions or potential applications of these techniques in industry or academia.
- Recommendation: Expand the conclusion to include specific future research directions and potential applications.
Thank you very much for your suggestions, and have reworked the Conclusions and Perspectives section to add Future Research Trends in Green Extraction Materials and Methods (lines 516-535) based on your suggestions.
- References:
- There is a need to assess the relevance and recency of references. Only two sources from 2022 are cited, which may not adequately reflect current research trends.
- Recommendation: Update the literature review to include more recent and relevant sources.
Thank you very much, the full text has been reorganized according to your suggestions, and new research results from recent years have been reintroduced and the citation section (lines 548-773) has been updated.
Reviewer 2 Report
Comments and Suggestions for Authors
This article reported the recent progress on green new phase extraction and preparation of polyphenols in edible oil. Some revisions should be conducted before consideration of publication.
1) The highlights should be added in the introduction section.
2) The comparison of deep eutectic solvent, ionic liquids and supercritical fluid should be added, and which compounds in edible oil were suitable?
3) The potential mechanism of extraction solvents should be added.
4) The tables about supercritical fluid extraction and supramolecular solvents should be summarized.
Comments on the Quality of English LanguageThis article reported the recent progress on green new phase extraction and preparation of polyphenols in edible oil. Some revisions should be conducted before consideration of publication.
1) The highlights should be added in the introduction section.
2) The comparison of deep eutectic solvent, ionic liquids and supercritical fluid should be added, and which compounds in edible oil were suitable?
3) The potential mechanism of extraction solvents should be added.
4) The tables about supercritical fluid extraction and supramolecular solvents should be summarized.
Author Response
Point-to-point responses
Thank you very much for giving us an opportunity to revise our manuscript. We have studied the comments. Any revisions have been marked in “Red” in revised manuscript. We appreciate for your warm work earnestly, and hope that the correction will meet with approval. Once again, thank you very much for your comments and suggestions. The point-to-point responses were listed as follows.
Reviewer #2
- The highlights should be added in the introduction section.
Thank you very much for your suggestions and have reworked the abstract section (lines 16-26) and the introduction section (lines 28-81) based on your suggestions. The logic of the introduction has been reorganized to more concisely state the objectives of the study.
- The comparison of deep eutectic solvent, ionic liquids and supercritical fluid should be added, and which compounds in edible oil were suitable?
Based on your suggestions have reworked Chapter 4, Application of Green Chemical Materials in Liquid Phase Extraction, by adding the following sections:
- comparison of green extraction solvents (lines 302-310);
- potential mechanism of DES (lines 319-323) and outlook (lines 358-363);
- suitability of ionic liquids for extraction of various compounds in edible oils (lines 409-411);
- The potential mechanism of extraction solvents should be added.
Based on your suggestions have added potential mechanisms of extraction solvents (lines 75-126) and added subheadings 2.1. Introduction to green extraction and 2.2. Application of green extraction technology, and improved the content.
In Chapter 4, Application of green chemical materials in liquid phase extraction, add limitations of green extraction solvents (lines 302-305) and potential mechanisms of DES (lines 319-323).
- The tables about supercritical fluid extraction and supramolecular solvents should be summarized.
Add table summarizing supercritical fluid extraction as per your suggestion, Table 3. Applications of supercritical fluids extraction (line 477). Add case studies and summaries to 4.4 Supramolecular solvents section (lines 506-512).
Round 2
Reviewer 2 Report
Comments and Suggestions for Authors
No comments